# Clinical Potential of Kinase Inhibitors in Combination with Immune Checkpoint Inhibitors for the Treatment of Solid Tumors

**DOI:** 10.3390/ijms22052608

**Published:** 2021-03-05

**Authors:** Ryuhjin Ahn, Josie Ursini-Siegel

**Affiliations:** 1Department of Biological Engineering, Koch Institute for Integrative Cancer Research, Massachusetts Institute of Technology, Cambridge, MA 02139, USA; ahnr@mit.edu; 2Department of Biochemistry, McGill University, Montréal, QC H3G 1Y6, Canada; 3Lady Davis Institute for Medical Research, Jewish General Hospital, Montréal, QC H3T 1E2, Canada; 4Department of Experimental Medicine, McGill University, Montréal, QC H3A 0G4, Canada; 5Department of Oncology, McGill University, 546 Pine Avenue West, Montréal, QC H2W 1S6, Canada

**Keywords:** kinase signaling, breast cancer, solid tumors, anti-tumor immunity, immunosuppression, kinase inhibitors, cancer immunotherapy

## Abstract

Oncogenic kinases contribute to immunosuppression and modulate the tumor microenvironment in solid tumors. Increasing evidence supports the fundamental role of oncogenic kinase signaling networks in coordinating immunosuppressive tumor microenvironments. This has led to numerous studies examining the efficacy of kinase inhibitors in inducing anti-tumor immune responses by increasing tumor immunogenicity. Kinase inhibitors are the second most common FDA-approved group of drugs that are deployed for cancer treatment. With few exceptions, they inevitably lead to intrinsic and/or acquired resistance, particularly in patients with metastatic disease when used as a monotherapy. On the other hand, cancer immunotherapies, including immune checkpoint inhibitors, have revolutionized cancer treatment for malignancies such as melanoma and lung cancer. However, key hurdles remain to successfully incorporate such therapies in the treatment of other solid cancers. Here, we review the recent literature on oncogenic kinases that regulate tumor immunogenicity, immune suppression, and anti-tumor immunity. Furthermore, we discuss current efforts in clinical trials that combine kinase inhibitors and immune checkpoint inhibitors to treat breast cancer and other solid tumors.

## 1. Introduction

The host immune system has multiple cellular machineries to eradicate malignant lesions. However, tumors develop multiple mechanisms to escape the host anti-tumor immune response. Immunotherapy, which re-engages immune surveillance pathways, has become one of the pillars of cancer treatment today. However, the ability of tumors to perpetuate an immunosuppressive microenvironment, combined with their ability to avoid being recognized as ‘non-self’, continues to impede the success of immunotherapy for many solid malignancies. Small-molecule kinase inhibitors represent an opportunity to overcome these key hurdles.

Over the past decades, studies have established that oncogenic kinases are fundamental in driving tumorigenesis and shaping the immune milieu to affect cancer progression and responsiveness to therapy. As of February 2021, at least 53 inhibitors (small-molecule or antibody-based) targeting more than 24 tyrosine/serine/threonine kinases have been approved by the FDA to treat various solid cancers. In line with the tumorigenic role of these kinases, many of these kinase inhibitors elicit anti-tumor immune responses, enhance tumor immunogenicity by regulating antigen processing and presentation and reduce immune suppression, which ultimately improves tumor killing. Many critical kinases are shared by tumor cells and immune cells, and the genetic or pharmacological inhibition of these kinases affects the function of both cell types. This has important implications in the clinical success of said inhibitors and warrants deeper understanding and attention as the research community explores ways to co-opt host immunity for cancer treatment.

Here, we provide a brief overview of the host immune responses mounted against malignant lesions, tumor immune evasion mechanisms, and challenges associated with immunotherapy and kinase inhibitors. We then review the mechanisms by which tumor-intrinsic oncogenic kinases shape the immune microenvironment, with a specific focus on the role of receptor and non-receptor kinases and their immediate molecular effectors. Preclinical studies and clinical trial results demonstrating that genetic and chemical perturbations of kinases can elicit anti-tumor immune responses to eradicate tumors, especially in combination with immunotherapy, are summarized. We spotlight on breast cancer as a model of recent advancements. Based on the evidence provided in this review, we posit that a rational, evidence-based combination of kinase inhibitors and immunotherapy may overcome some of the hurdles faced by both therapeutic modalities and improve the treatment of cancer patients.

## 2. Cancer Immunosuppression and Anti-Tumor Immunity

The true appreciation of the immune response in suppressing tumor formation came when mice lacking adaptive immunity (RAG2 knock-out) showed an increased tumor incidence upon carcinogen exposure [1]. Numerous immune cell types collaboratively mediate tumor cell killing through multiple mechanisms, including the recognition of tumor-associated or tumor-specific antigens by the adaptive immune system and non-antigen-dependent killing by the innate immune system. Cytotoxic T lymphocytes (CD8+ CTL) are part of the adaptive immune system and induce tumor cell apoptosis through (1) recognition of antigens presented in the context of MHC class I leading to perforin/granzyme B secretion and (2) recognition of Fas on tumors via FasL, leading to caspase-mediated cell death. CTLs also release cytokines such as TNFα and IFNγ to promote cell cycle arrest in tumors [2,3,4,5,6]. While CTLs mount their attacks via T cell receptor (TCR) mediated recognition of antigen-MHC complexes, natural killer (NK) cells induce similar cytotoxic responses, but in an antigen-independent manner. Instead, the cytotoxic potential of NK cells is controlled by the net balance of stimulatory versus inhibitory receptors on the surface of NK cells themselves, combined with particular classes of ligands expressed by the target cells [7,8]. In oncology, immunogenic cell death (ICD) represents a type of non-microbial death that can be initiated, either due to endoplasmic reticulum stress or in response to cytotoxic treatments, such as anthracyclines or radiotherapy. These various stressors lead to the release of damage-associated molecular patterns (DAMPs), which prime the innate immune system to educate potent anti-tumor immune responses. B cells of the adaptive immune system have also emerged as playing anti-tumorigenic roles through (1) release of tumor antigen-specific antibodies (once B cells differentiate into plasma cells) that trigger antibody-dependent cellular cytotoxicity (ADCC) by NK cells or complement-dependent cytotoxicity (CDC), and (2) B cell receptor-mediated antigen presentation to CD8+ or naïve CD4+ cells for tumor killing [9,10,11]. During ADCC, membrane-bound antigens on the surface of tumor cells are recognized by specific antibodies. NK cells expressing Fc receptors then bind the Fc portion of these antibodies, leading to NK cell activation and subsequent release of cytotoxic granules that elicit tumor cell lysis [12]. Finally, beyond immune cells with direct tumoricidal properties, infiltrating dendritic cells into tumors are central for activating CTL-driven anti-tumor immunity, as professional antigen-presenting cells that educate both Th and CTL immune responses (reviewed in [13]).

The ability of malignant cells to escape from such anti-tumor immune responses through promoting immunosuppression has been established as a critical hallmark of cancer [14]. Malignant cells achieve this by organizing into a complex structure composed of diverse cell types, including stromal cells, immune cells, and endothelial cells, all of which are in constant communication [15]. Tumors develop multiple tiers of immunosuppressive mechanisms to escape the host immune response, which has been extensively reviewed [16,17]. Briefly, this involves (1) secretion of immunosuppressive cytokines that inhibit anti-tumor adaptive (e.g., CTLs) and innate immune cells (e.g., NK cells) and polarize immune cells to pro-tumorigenic subtypes (e.g., T regulatory cells; T_reg_), (2) secretion of chemokines that recruit immunosuppressive stromal and immune cells (e.g., myeloid-derived suppressor cells, tumor-associated fibroblasts, and macrophages) that in turn secrete immunosuppressive cytokines (e.g., IL-10, TGFβ), (3) promotion of anergy and tolerance in anti-tumor immune cells through expression of surface inhibitory ligands (e.g., PD-L1) and persistent self-antigen presentation, (4) suppression of antigen presentation through, e.g., epigenetic mechanisms [18], to avoid detection by adaptive immunity and (5) upregulation of signaling pathways that reduce necessary metabolites (e.g., ATP) for immune cell (e.g., immature DC) activation in the tumor microenvironment [16].

Tumors can be classified as (1) immune cold (lack of immune infiltration) due to a lack of tumor antigens, deficiency in antigen presentation, absence of T cell priming, and impaired T cell trafficking, or (2) immune hot (high immune infiltration and increased mutational burden providing an abundance of tumor-specific neoantigens) [19]. Furthermore, not only the number of tumor-infiltrating lymphocytes (TIL; especially CD8+ T cells) but their spatial organization contribute to prognostic and predictive stratification in breast cancer [20,21,22,23,24]. This is also observed in ovarian cancer [25] and early-stage non-small cell lung cancer [26]. Thus, spatial, quantitative, and qualitative differences in the type of immune infiltrates in the tumor microenvironment are prognostic of disease outcome.

## 3. Cancer Immunotherapy and Challenges

In recent years, cancer immunotherapy has revolutionized the treatment of cancer. It aims to reinstate immune surveillance, turning cold tumors into hot tumors to eradicate cancer [19]. Numerous cancer immunotherapy modalities have been developed, and they are extensively reviewed [27]. Many solid tumors establish immune suppression by upregulating the expression of key immune checkpoint receptors (e.g., PD-1, CTLA4) on infiltrating T cells as well as immunosuppressive ligands (e.g., PD-L1, PD-L2, B7-H4), either on tumor cells or other cell types in the tumor microenvironment. [28,29]. Normally, PD-1 expressed on T cells restricts peripheral tissue damage and inflammation through limiting TCR-mediated effector T cell function and maintaining self-tolerance [29]. Tumors co-opt this pathway and promote immunosuppression by expressing PD-L1 downstream of oncogenic signaling pathways (e.g., STAT3, STAT1, Myc, 9p24.1 amplification, CD44 [30]) or following exposure to IFNγ secreted by T cells [29].

In this regard, the advent of immune checkpoint inhibitors (ICIs) targeting either CTLA-4, PD-1, or PD-L1 has revolutionized the field of cancer therapy, resulting in sustained clinical remissions in patients otherwise refractory to standard of cancer therapies in many cancer types, including melanomas and lung cancers. Indeed, there are currently one CTLA-4 and six PD-1/PD-L1 inhibitors approved across ten tumor types and numerous stages of cancer [31]. Unfortunately, while checkpoint inhibitors have resulted in strong and durable responses in some cancers, a large proportion of tumor types remain refractory to this treatment modality [32]. While a high mutational burden and increased density of infiltrating TILs are often predictive or response to ICIs, key questions remain. First, what other cell types dictate immunotherapeutic responsiveness (e.g., composition and the landscape of immune cells in the tumor microenvironment, unique tumor-specific signaling mechanisms) or lack thereof? Second, what are the predictive biomarkers for therapeutic responsiveness beyond the drug target itself? Third, what are the resistance mechanisms, conferred either by tumor cells or those within the local microenvironment? Based on the molecular understanding of how tumor-intrinsic signaling alters the immune response, a rational combination of chemotherapy, targeted therapy, and cancer immunotherapy methods need to be explored. Indeed, chemotherapy has already been shown to improve the sensitivity of immune checkpoint inhibitors (as reviewed in [33]).

In breast cancer, immune checkpoint inhibitors as single-agent or in combination with other therapies have been explored [34,35]. Stanton et al. examined 13,914 patient samples and determined that 5–26% of breast cancers have high infiltration of lymphocytes while 16% of cancers showed no infiltration. A median of 20% triple-negative (TNBC), 16% HER2+, and 6% ER+/PR+/HER2- (HR+) breast cancers show predominant lymphocyte infiltration (defined as >50% lymphocytic infiltrate) [36]. CD8+ CTLs indicative of anti-tumor immune responses, as well as FOXP3+ T_reg_ cells indicative of a tumorigenic immune response, were most prominent in TNBC (60% infiltrated with CTLs and 70% infiltrated with T_reg_) and HER2+ (61% and 67%) tumors compared to HR+ breast cancers (43% and 38%) [36]. These data indicate that subsets of breast cancers, especially within the TNBC and HER2+ subtypes, are relatively more immunogenic and contain high TILs [37,38]. Importantly, high infiltration of TILs has been associated with improved prognosis in early-stage TNBCs and HER2+ breast cancers [21,39,40,41], while the opposite is true in luminal breast cancer [42]. Based on these findings, immunotherapy approaches have been explored mostly in TNBC and HER2+ subtypes. As of January 2021, two immune checkpoint inhibitors have been approved by the FDA for breast cancer: (1) atezolizumab (PD-L1 inhibitor) with protein-bound paclitaxel for locally advanced, non-removable TNBC or metastatic TNBCs that are PD-L1 positive (IMpassion130 trial) [43], and (2) pembrolizumab (PD-1 inhibitor) for locally recurrent unresectable or metastatic TNBCs that are PD-L1 positive (KEYNOTE-355) [44]. Several important insights have been made from these and other preclinical studies. First, tumor mutational burden predicts prolonged survival associated with high (and not low) immune infiltration in breast cancer [45], especially in TNBC and HER2+ subtypes [34]. This is consistent with observations made in lung cancer and melanoma [46]. Second, only a subset of metastatic breast cancer patients, especially those expressing high tumor PD-L1, benefit from immune checkpoint blockade therapy [47]. Third, multiple layers of immunosuppressive mechanisms exist in the tumor microenvironment to impede treatment responsiveness [48]. Indeed, the single-agent activity of avelumab (PD-L1 inhibitor) [49], atezolizumab [50], and pembrolizumab [51] have shown limited activity in breast cancer patients, underscoring the need for rationally designed combination approaches [35].

## 4. Tumor-Intrinsic Kinase Signaling That Coordinate Cancer Immunosuppression

Since the proposal that cancer is a wound that never heals [52], studies in the 1990s demonstrated that tumors co-opt inflammation for survival [53,54,55,56]. Malignant progression not only relies on intrinsic signaling (loss of tumor suppressors and gain of oncogenes due to genetic aberrations) but also on extrinsic cellular players from the local microenvironment [57]. Today, it is fully established that tumor cell-intrinsic mechanisms continuously shape the tumor immune landscape to favor cancer progression and therapeutic resistance [57,58]. Oncogenic receptor and non-receptor kinases are crucial contributors of tumor-intrinsic signaling that coordinate cancer immunosuppression through diverse mechanisms (Figure 1).

## 5. Non-Receptor Kinases in Immunosuppression

One of the first demonstrations of an oncogenic kinase directly impacting the immune landscape came in 2004, whereby the H-RasG12V oncogene was shown to induce *CXCL8* transcription in various cancer cell lines to promote macrophage infiltration and vascularization in vivo [59]. Numerous other studies have further linked deregulated activation of serine/threonine kinases, including the Ras/Raf/Mek/Erk and PI3K/AKT pathways as well as cyclin-dependent kinases, to the establishment of chronic inflammation in solid tumors. For example, BRAFV600E, a constitutively active form of the BRAF serine/threonine kinase, drives melanoma and has been shown to promote IL-6, IL-10, and VEGF secretion in a STAT3 dependent manner [60]. This, in turn, could suppress LPS induced inflammation by dendritic cells. Indeed, combination therapies that include MEK inhibitors with immunostimulatory agonists induce profound immunogenic responses in preclinical models of K-Ras positive pancreatic cancer by limiting the activation of immunosuppressive cell subsets, including M2 macrophages, myeloid-derived suppressor cells, and regulatory T cells [61]. In melanoma, the loss of PTEN (tumor suppressor that negatively regulates AKT/PI3K activity) leads to decreased numbers of TILs, reduced responsiveness to PD-1 checkpoint inhibition, and increased secretion of immunosuppressive cytokines [62]. In breast cancer patients, inhibition of cyclin-dependent kinases 4 and 6 (CDK4/6), which are fundamental drivers of cell cycle progression downstream of oncogenic signaling pathways, induces significant anti-tumor immune responses [63]. Non-receptor tyrosine kinases also contribute to cancer immune suppression. Indeed, nasopharyngeal carcinoma cells secrete ISG15, an IFN-responsive gene, which activates Src family kinases in macrophages to promote their M2 polarization and immunosuppressive properties [64]. In squamous cell carcinoma, nuclear focal adhesion kinase (FAK) regulates transcription of CCL5 to promote T regulatory cell recruitment and exhaustion of CD8+ T cells to promote tumor growth, and treatment with FAK inhibitors reactivates anti-tumor immune responses [65]. Together, these studies suggest that targeting non-receptor kinase signaling networks may represent a promising therapeutic strategy to relieve tumor immune suppression, particularly as part of a rationally designed combination therapy.

## 6. Receptor Tyrosine Kinases in Immunosuppression

Overexpression or activating mutations in receptor tyrosine kinases leading to their aberrant activation are critical drivers of various solid tumors [66]. Receptor tyrosine kinases (RTKs) are single-pass transmembrane proteins expressed on the surface of various cell types that regulate proliferation, differentiation, survival, metabolism, migration, and invasion in cancer [66]. There are 58 known RTKs that fall into 20 subfamilies [66]. Upon binding to their cognate ligands, RTKs undergo conformational changes, leading to receptor dimerization and activation of their tyrosine kinase domains. These activated kinases then trans-phosphorylate key tyrosine residues in their cytoplasmic tails, leading to the recruitment of adaptor proteins that initiate downstream signaling pathways such as the mitogen-activated protein kinase (MAPK) and phosphoinositide 3- kinase (PI3K)/AKT pathways [67].

Studies have reported essential roles for several RTKs, including CSF1R, VEGFR, RON, and the TAM family, in promoting cancer immune suppression (reviewed in [68,69,70,71]). More recent studies have further elucidated the prominent examples of EGFR, HER2, and AXL signaling contributing directly to tumor-driven immunosuppression. For example, the high activation status of EGFR and HER2 is associated with increased PD-L1 expression in gastric cancer cells and patient tumor tissues [72]. Indeed, EGFR signaling in lung cancer activates the PD-1 immune checkpoint to promote immune evasion [73]. Constitutively active EGFR variants can induce immune suppression in lung cancers through their ability to shed mutant EGFR-containing exosomes into infiltrating dendritic cells, abrogating their ability to present tumor antigens [74]. Finally, increased Axl signaling stimulates NF-κB signaling to potentiate chronic inflammation and subsequent immune evasion in pancreatic cancer cells [75]. Together, these studies highlight an emerging role for RTKs in the establishment and maintenance of immune suppression and highlight the therapeutic potential for combining tyrosine kinase inhibitors with immunomodulatory agents (as discussed below).

## 7. Adaptor Proteins in Immunosuppression

Adaptor proteins are critical integrators of downstream tyrosine kinases to initiate oncogenic signaling cascades. They do so by nucleating signaling complexes through their ability to engage in both phospho-tyrosine dependent and –independent interactions. For example, the Shc1 adaptor protein is recruited to activated oncogenic RTKs and tyrosine kinases (TKs), such as EGFR, ERBB2, ERBB3, ERBB4, INSR, IGF1R, VEGFR3, FGFR1, TrkA, RET, MET, FGFR2, VEGFR2, c-kit, JAK3, EphA2, Src family kinases, Alk, PDGFRa, Ron, TrkB, and Axl through phospho-tyrosine binding domains such as PTB or SH2 domains [76,77,78,79,80,81,82,83]. This subsequently allows Shc1 to become phosphorylated on key tyrosine residues (Y239/240 and Y317, equivalent to Y313 in mice), which nucleates downstream signaling complexes to activate PI3K/AKT and Ras/MAPK oncogenic pathways [84]. Indeed, Shc1 signaling downstream of ErbB2 and polyomavirus middle T antigen (PyMT) oncogenes leads to suppression of CTL infiltration and IFNγ-driven immunity during the early stages of mammary tumorigenesis [85]. Importantly, mammary tumor progression of Shc1 deficient hyperplasias is significantly accelerated in athymic mice compared to immunocompetent animals, suggesting that Shc1 signaling suppresses T cell immune responses, ultimately facilitating tumor progression [85]. This is partially mediated by tyrosine 239/240 and tyrosine 313 residues of Shc1, which activate STAT3-dependent immunosuppression and inhibit STAT1-induced immune surveillance in breast cancer cells, respectively [86]. Loss of phospho-Y313-Shc1 signaling is associated with a STAT1 dependent increase of MHC class I surface expression. It is plausible that numerous tumor intrinsic TKs may be implicated in regulating immunosuppression through Shc1. In line with this, targeting the pathways closely upstream or downstream of Shc1 (e.g., ErbB2, EGFR, MEK, ERK, PTEN, PI3K) by drugs or molecular manipulation has been shown to enhance anti-tumor immune responses in various studies, as discussed later. Moreover, Crk, another adaptor protein primarily known for its role in cell adhesion and growth factor signaling, has been demonstrated to promote immunosuppression in the 4T1 murine model of breast cancer [87]. Deletion of Crk enhances anti-tumor immune responses and secretion of cytokines that favor immune surveillance, leading to reduced tumor growth and metastasis. Loss of Crk also enhanced tumor clearance upon PD-1 checkpoint inhibition [87]. Taken together, these studies demonstrate the immunosuppressive role of oncogenic signaling pathways and how they may be targeted to elicit anti-tumor immunity.

## 8. Kinase Inhibitors Potentiate the Tumoricidal Responses of Immunotherapy

Oncogenic kinases drive immunosuppression, and their inhibition elicits anti-tumor immune responses, yet with inevitable resistance and recurrence. Despite the success of immunotherapy in oncology, only select patients experience durable responses due, in part, to persistent immunosuppression in the tumor microenvironment. Thus, a concomitant reduction of immunosuppression by kinase inhibitors while unlocking the therapeutic potential of tumoricidal immune cells has gained attention as a strategy to extend the clinical benefit of the current standard of therapies. To this end, roughly 53 small molecule kinase inhibitors and 12 antibody-based kinase inhibitors have been approved by the FDA for cancer treatment, 11 of which are for being studied in breast cancer (Appendix A). Kinase inhibitors have provided significant improvement in clinical outcomes in various cancers. However, they often lead to acquired resistance and recurrence after long-term exposure due to functional redundancy of the kinome and emergence of mutant variants that are resistant to the kinase inhibitor. Clinical trials have been initiated for various cancers to evaluate the efficacy of kinase inhibitors in combination with cancer immunotherapy (summarized in Appendix A).

## 9. Tyrosine Kinase Inhibitors

Cancer cells rely on a large family of cytoplasmic and receptor tyrosine kinases to initiate tumorigenic signaling pathways that ultimately activate downstream signaling molecules, including serine/threonine kinases, which are effectors of their immunosuppressive properties. In particular, pharmacological inhibitors targeting the ErbB2 RTK family (EGFR, ErbB2, ErbB3, and ErbB4) are standard of care for subsets of breast cancer and NSCLC patients and have been shown to elicit potent anti-tumor immune responses in the tumor microenvironment, thereby increasing the clinical impact of immune checkpoint inhibitors (Appendix A and as reviewed in [88]). Several other tyrosine kinase inhibitors (TKI) also induce CTL-driven anti-tumor immune responses, including sunitinib, a broad spectrum RTK inhibitor that targets VEGFR, PDGFRα, Ret, and Kit, which has been shown to reverse immune suppression by inhibiting STAT3 signaling in renal cell carcinoma [89]. BMS-777607/ASLAN002, a Ron-selective kinase inhibitor, reduces breast cancer lung metastasis of breast cancer by promoting anti-tumor immune responses [90].

Recent studies have further elucidated the mechanisms of action by which small-molecule tyrosine kinase inhibitors targeting the ErbB2 family evokes tumoricidal immune responses. The treatment of EGFR and HER2 overexpressing gastric cancer cells with Afatinib (a pan-ErbB2 family TKI) and lapatinib (EGFR, ErbB2 inhibitor) reduced PD-L1 expression levels [72]. Lapatinib treatment further decreased the secretion of immunosuppressive cytokines (e.g., CCL2, VEGF) from HER2-amplified tumor cells [72]. An ErbB2-driven breast cancer mouse model (MMTV/Neu) also responded to lapatinib treatment with increased IFNγ driven anti-tumor adaptive immune responses in a STAT1 dependent manner [91].

Other solid tumors undergoing clinical trials to test the efficacy of kinase inhibitors in combination with immunotherapy include non-small cell lung cancer [92], lung squamous cell carcinoma [93], and advanced renal cell carcinoma [94,95], and breast cancer (Appendix A). In hepatocellular carcinoma, immunotherapy has shown efficacy in the second-line setting, while tyrosine kinase inhibitors have shown benefit both in the first- (Sorafenib) and second-line (regorafenib, cabozantinib, and ramucirumab) settings [96,97,98]. Based on this, preliminary trials are ongoing in hepatocellular carcinoma patients to determine how the combination of tyrosine kinase inhibitors and immunotherapy prolongs survival compared to each as a single agent [99]. Additionally, promising safety profiles and results were seen in NSCLC (erlotinib and nivolumab) [100], while high toxicities have halted some trials [101]. In May 2019, a breakthrough randomized clinical trial in renal cell carcinoma (JAVELIN Renal 101) showed avelumab in combination with axitinib resulted in a significant survival benefit compared to standard of care sunitinib, leading to the FDA approval of this regimen [102].

Monoclonal antibody-based therapies targeting ErbB2 family signaling are also employed in cancer patients and rely, in part, on an intact immune system to achieve maximal clinical benefit. EGFR-neutralizing antibodies combined with chemotherapy depend on immunogenic cell death (ICD) to clear colorectal cancers [103]. Trastuzumab, a recombinant humanized monoclonal antibody directed against the human HER2 receptor tyrosine kinase, mediates tumor-killing partially by inducing antibody-dependent cell-mediated cytotoxicity (ADCC) against HER2 overexpressing tumor cells. Accordingly, the therapeutic effect of trastuzumab is diminished in mice that lack NK cells or those that have macrophages disabled to bind the Fc region of trastuzumab [104]. Similarly, NK cell-derived IFNγ induced PD-L1 expression in tumors and enhanced cetuximab (EGFR inhibitor)-mediated ADCC [105].

Finally, evidence suggests that cross-talk between cancer cells and immune cells influences the therapeutic responsiveness of these targeted therapies. One high-throughput immuno-oncology screen identified the EGFR inhibitor erlotinib as a potent enhancer of antigen-specific CTL tumor cell killing, synergizing with anti-PD-1 checkpoint inhibition to suppress colon cancer growth [106]. Preclinical studies in lung cancer show that the EGFR pathway enhances immunosuppression through increased engagement of PD-1/PD-L1 and CTLA4 in an ERK and NFkB dependent manner [107]. In line with this, erlotinib treatment in transgenic mice that develop EGFR^L858R^-driven lung cancers induced the infiltration of T cells, B cells, NK cells, and CD11c+MHC-II+ cells as well as immunosuppressive CD11b+Gr1+ MDSCs [108]. Unlike the initial anti-tumorigenic immune infiltrates that are induced by EGFR TKIs, these therapy-induced increases in MDSC populations persist and are accompanied by increased levels of circulating immunosuppressive cytokines (IL-10, CCL2) in serum [109]. Thus, identifying strategies to prevent and/or overcome such TKI-induced immunosuppressive adaptive responses are required to achieve durable clinical benefit with these classes of drugs, particularly in combination with immune checkpoint inhibitors.

## 10. Serine/Threonine Kinase Inhibitors

Numerous signal transduction pathways, downstream of tyrosine kinases, bifurcate on key serine/threonine kinases, which phosphorylate effector molecules to potentiate the emergence of aggressive cancers and the establishment of an immunosuppressive tumor microenvironment.

### 10.1. CDK4/6 Inhibitors

This family of inhibitors, including abemaciclib, palbociclib, and ribociclib, targets CDK4 and CDK6, two serine/threonine kinases that are required for cell cycle progression. There are currently over 200 clinical trials ongoing with CDK4/6 inhibitors across multiple tumor types, including breast, lung, ovarian, colorectal, and prostate cancers. These inhibitors have shown particular promise in treating metastatic breast cancers, particularly in ER-positive tumors in combination with hormonal therapies. Indeed, abemaciclib has been approved by the FDA for the treatment of advanced HR+HER2- breast cancer, based on the MONARCH 3 trial [110]. Abemaciclib, in combination with anastrozole (an aromatase inhibitor), results in increased adaptive immune response signatures that are phenotypic of increased T cell activation and antigen presentation even in early-stage HR+HER2- breast cancer, providing optimism for ongoing clinical trials in HR2+HER2- disease (Appendix A) [111]. Preclinical studies further show that CDK4/6 inhibitors delay colorectal cancer growth in syngeneic mice, in part, by stimulating tumoricidal immune responses. This includes increased T cell infiltration, T cell effector function, antigen processing and presentation, macrophage and dendritic cell activation, combined with a potent inflammatory response [112,113]. Indeed, abemaciclib further potentiated the therapeutic benefit of PD-L1 inhibitors in colorectal tumors [112]. Unexpectedly, however, CDK4/6 inhibition increased PD-L1 protein stability in tumor cells by preventing proteasome-mediated PD-L1 degradation [114]. These studies highlight the mechanistic basis for combining CDK4/6 inhibitors and PD-1/PD-L1 inhibitors as this combinatorial approach stimulates an immunogenic tumor microenvironment and primes cancers for PD1/PD-L1 immune checkpoint blockade.

### 10.2. RAF/MEK Inhibitors

There has been significant interest in exploring RAF/MEK inhibitors as enhancers of anti-tumor immune responses based on early preclinical studies showing that MEK inhibitors potentiate CD8+ T cell responses by preventing an exhausted phenotype in cancer models, leading to durable responses in combination with PD-L1 inhibitors [115]. Vemurafenib and dabrafenib are two selective V600 mutant BRAF inhibitors that have been deployed for the treatment of V600E+ tumors, including melanoma, colorectal cancer, and non-small cell lung cancers. Cobimetinib and trametinib are MEK1/2 inhibitors that function downstream of BRAF and are indicated for treating both BRAF wild-type and mutant tumors. Numerous clinical trials have reported on the efficacy of this family of inhibitors in overcoming immune suppression. Recent clinical trials have evaluated the safety and efficacy of triple therapies that combine a BRAF and MEK inhibitor together with an immune checkpoint inhibitor. For example, a recent phase Ib study tested vemurafenib and cobimetinib, combined with a PD-L1 inhibitor (atezolizumab), in patients with BRAF V600E metastatic melanoma (NCT1656642). Durable anti-tumor responses were observed in 40% of patients even 30 months following combination treatment, which was associated with increased recruitment of activated TILs [116]. This is further supported by a recent phase 3 COMBI-I clinical trial (NCT02967692) that examined the efficacy of a novel PD-1 inhibitor (spartalizumab) in combination with dabrafenib and trametinib in patients with unresectable and/or metastatic BRAF V600E mutant melanomas. A 78% objective response rate was observed, whereby 44% of patients showed a complete pathological response. Although 72% of patients experienced an immune-related adverse event, this study suggested that combined inhibition of BRAF and MEK signaling together could increase the efficacy of immune checkpoint inhibitors [117]. Preclinical studies in mouse models of BRAF V600E mutant melanoma have begun to elucidate the mechanistic basis for improved immunological responses to combined treatment with BRAF and MEK inhibitors. Combined dabrafenib and trametinib treatment increased the infiltration and cytotoxicity of TILs, which was associated with decreased recruitment of tumor-associated macrophages, T regulatory cells and increased antigen processing and presentation of melanosomal antigens [118]. Trametinib has been associated with increased HLA expression, increased CD8+ T cell infiltration, and improved tumor control in combination with checkpoint inhibitors [119,120,121,122]. In addition, dual BRAF/MEK inhibition further decreased the production of immunosuppressive adenosine in BRAF mutant melanoma cells by reducing the expression of components within the CD73 adenosinergic pathway [123].

### 10.3. PI3K/mTOR Inhibitors

The PI3K/AKT/mTOR signaling pathway is commonly activated in solid tumors and is critical for tumor progression, metastatic spread, and therapeutic resistance [124,125]. PI3K proteins are phosphatidylinositol 3′ lipid kinases and are composed of heterodimers with regulatory (p85) and catalytic subunits (p110). Four catalytic isoforms (p110α, p110β, p110γ, p110δ) are activated downstream of tyrosine kinases, and G-protein coupled receptors, leading to the activation of multiple serine/threonine kinases, including several AKT family members (AKT1, AKT2, AKT3) and mechanistic target of rapamycin (mTOR), which functions within two distinct protein complexes (mTORC1 and mTORC2). Together, these serine/threonine kinases phosphorylate hundreds of target proteins with oncogenic functions. Indeed, PI3K/AKT/mTOR functions as a nutrient sensor to induce cancer cell proliferation and survival under nutrient replete conditions [126]. It does so, in part, by increasing the rate of protein synthesis by promoting the assembly of the eIF4F complex (eIF4E/4A/4G), which potentiates cap-dependent translation initiation. Indeed, mRNA translation of genes with oncogenic properties is selectively controlled by eIF4E availability, the rate-limiting step for eIF4F complex assembly. Moreover, phosphorylation of eIF4E by MNK1/2, another class of serine/threonine kinases, further potentiates the rate of mRNA translation of eIF4E-sensitive genes [127].

Dysregulation of the PI3K/AKT/mTOR signaling pathway in solid tumors has been implicated in the establishment of immunosuppression by numerous studies by stimulating the production of immunosuppressive cytokines, recruitment of immunosuppressive cell types, and induction of immune checkpoint ligands on tumor cells (as reviewed in [128,129]). Recent literature suggests that PTEN-deficient tumors show increased expression of immunosuppressive cytokines and genetic silencing of PTEN expression in melanoma cells attenuated anti-tumorigenic T cell responses *in vivo*, leading to resistance to PD-1 immune checkpoint blockade [62]. Intriguingly, another study found that PTEN was essential to induce IFN-driven innate immunity and potentiate anti-viral responses independent of its role in negatively regulating PI3K/AKT signaling. Instead, PTEN protein phosphatase activity was shown to control IRF3 nuclear import, a key transcription factor that regulates type I IFN responses [130]. This study highlights potential novel and unappreciated roles for PTEN in promoting immune surveillance in solid tumors. Having said this, other preclinical studies clearly suggest the mTOR/MNK/eIF4E activation in solid tumors promotes an immunosuppressive microenvironment. For example, mTOR signaling in breast cancer cells increases G-CSF expression to stimulate the recruitment of immunosuppressive MDSCs [131]. Deregulated PI3K signaling in tumor cells further establishes an immunosuppressive niche by inducing activation of pro-inflammatory mediators, including nitric oxide synthase and lipoxygenase, in the tumor microenvironment [132]. Furthermore, drugs targeting the mRNA translation machinery, including eFT508 (MNK1/2 inhibitor) and silvestrol (eIF4A inhibitor), reduce PD-L1 expression by tumor cells in models of liver cancer and melanoma, sensitizing tumors to T cell-dependent immune responses [133,134]. Finally, early preclinical studies suggested that combining a pan-PI3K inhibitor with an immune adjuvant induces production of IFNγ and IL-17-producing inflammatory T cells, leading to profound anti-tumorigenic immune responses in mouse models of lung cancer and melanoma, paving the way for future studies looking at combination therapy with immunotherapies [135].

In light of these observations, significant efforts by the pharmaceutical sector have yielded numerous inhibitors that selectively target PI3K/AKT/mTOR signaling [136,137]. These include pan-PI3K and/or dual PI3K/mTOR inhibitors (buparlisib, bimiralisib, copanlisib, dactolisib, idelalisib, apitolisib, gedatolisib, tenalisib), isoform-specific PI3K inhibitors (taselisib, alpelisib, parsaclisib, serabelisib, umbralisib), AKT inhibitors (AZD5363, MK2206), mTORC1-selective inhibitors (rapalogs, including temsirolimus, everolimus), dual mTORC1/mTORC2 inhibitors (asTORi, including INK128, AZD8055, LXI-15029) and MNK inhibitors (tomisvosertib). Several recent phase I clinical trials with these inhibitors demonstrate their therapeutic potential in reversing immune suppression. For example, a phase I study testing taselisib, a p110α-specific inhibitor, in women with triple-negative breast cancer showed increased expression of genes associated with activated T cell and NK cell responses, co-incident with anti-tumor responses [138]. Moreover, SF2523, a pan-PI3K and dual BRD4 inhibitor, inhibited the growth of lung, melanoma, and colorectal cancers in syngeneic models, which was associated with reduced infiltration of MDSCs and restoration of CD8+ T cell function [139]. Indeed, in murine models of metastatic breast cancer, buparlisib induced an inflammatory response and synergized with PD-1 neutralizing antibodies [140]. Temsirolimus, an mTOR inhibitor, could enhance anti-tumor immune responses in melanoma and renal cell cancer mouse models when used with cancer vaccines [141]. In contrast, everolimus, a mTORC1-specific inhibitor, has been shown to suppresses CTL and NK cell function and upregulate the presence of regulatory T cells [142,143], which is in line with its use as an immunosuppressant in organ transplantation [144,145]. These negative effects could be alleviated by combining cyclophosphamide with everolimus, leading to the depletion of Tregs and MDSCs, co-incident with an increased level of CD8+ effector T cells in the blood of patients with metastatic renal cell carcinoma [146]. This highlights the need for careful consideration into drug combinations. Together, these studies establish that pharmacological targeting of the PI3K/AKT/mTOR signaling pathway may synergize with immune checkpoint inhibitors in eliciting tumoricidal immune responses in solid tumors.

## 11. Combination Strategies Targeting Kinase Inhibitors Improve the Efficacy of Immune Checkpoint Inhibitors in Breast Cancer

Durable clinical responses to immune checkpoint inhibitors have only been observed in cancer types, including melanoma and NSCLCs, which exhibit a high degree of genomic instability and are immunologically “hot” tumors. In contrast, most breast cancers are not infiltrated by abundant TILs and have low levels of microsatellite instability, resulting in the paucity of available tumor antigens. In this regard, clinical trials examining whether rationally designed combination therapies can increase immunological responses are ongoing. Many breast cancer clinical trials using immunotherapy combined with chemotherapy or other targeted agents have mostly been in TNBC and HER2+ tumors as these subtypes display the highest immunogenicity (PD-L1+ TIL, PD-L1+ tumor, mutation, neoantigen load, and MHC expression) [34]. Other kinase inhibitors that have shown promise include Raf/MEK inhibitors, PI3K/mTOR inhibitors, CDK4/6 inhibitors, and HER2 inhibitors [35]. Current clinical trials combining kinase inhibitors with checkpoint inhibitors in breast cancer are summarized (Appendix A). Biomarkers associated with immunomodulation are being assessed with a renewed focus for dual treatment of lapatinib and trastuzumab (NCT02213042) or trastuzumab and pertuzumab (NCT03144947) in the neoadjuvant setting in advanced HER2+ breast cancer patients, which may further guide future combination strategies.

Beyond checkpoint inhibitors, cancer vaccines have also been tested in breast cancer clinical trials, albeit with limited success. Metastatic, trastuzumab-refractory HER2+ breast cancer patients were treated with lapatinib and a HER2-based cancer vaccine (a recombinant protein with extracellular domain and part of the intracellular domain of HER2 combined with an adjuvant) concurrently based on success in a preclinical model. However, no objective clinical responses were seen [147]. Previously, a single-arm, non-randomized feasibility study in HER2+ metastatic breast cancer patients (*n* = 20) was done using a HER2+ whole-cell breast cancer vaccine and weekly trastuzumab. This showed a 6-month clinical benefit of 55%, which was supported by mouse model studies with control groups [148]. While the results were encouraging, further studies with larger cohorts and control arms are necessary to determine the true benefit of cancer vaccines to treat HER2+ breast cancers.

## 12. Kinase Inhibitors and Tumor-Intrinsic Antigen Processing and Presentation

Intriguingly, multiple studies have linked the efficacy of kinase inhibitors to high MHC class I antigen presentation by tumor cells. Cabozantinib (targets RET and MET) has been shown to increase MHC class I (H-2Db) and Fas expression in colon cancer cell lines [149]. In NSCLC, mutation or overexpression of EGFR promotes immunosuppression, and the inhibition of EGFR using gefitinib or erlotinib can restore MHC class I expression, reduce PD-L1 expression or upregulate the expression of NKG2D ligands for NK cell-mediated tumor killing [107]. Furthermore, CDK4/6 inhibitor abemaciclib induces upregulation of antigen presentation in the context of MHC class I, leading to breast tumor regression [63,112]. A similar increase in tumor cell surface MHC-I expression upon abemaciclib treatment was observed in a mouse colorectal tumor model [112] and RB positive Ewing sarcoma preclinical model [113]. BRAF inhibitor vemurafenib has been shown to upregulate MHC in BRAFV600E homozygous melanoma [150]. High-throughput shRNA screens revealed that MEK, EGFR, and RET negatively regulate antigen processing and presentation machinery and MHC class I expression in an ERK-dependent manner [151]. In line with this, pharmacological inhibition of these kinases led to improved T cell-mediated killing through antigen-MHC recognition [151]. FDA-approved tyrosine kinases that alter antigen presentation pathways are noted in Appendix A. It was recently shown that palbociclib (CDK4/6 inhibitor), in addition to increasing the MHC class I expression, can alter the peptide-MHC repertoire in melanoma cell lines to reflect the intracellular response to CDK4/6 inhibition [152]. How the quality and quantity of tumor-associated antigen repertoires are impacted by kinase inhibitors will have important implications for therapeutic cancer vaccine development and other cell-based immunotherapy modalities that target tumor-specific antigens [153]. Together, these studies provide the basis for a strategic combination of kinase inhibitors (small molecule inhibitor and antibody-based) with synthetic immune-based therapy (engineered TCR-based or antibody-based) for the treatment of cancer.

## 13. Kinase Inhibitors Impact JAK/STAT-Mediated Tumor Immunity

The JAK/STAT signaling pathway is critical for the initiation and subsequent resolution of inflammatory responses. Janus kinases (JAK) are a family of tyrosine kinases (JAK1, JAK2, JAK3, and TYK2) that are activated downstream of multiple cytokine receptors, both in tumor cells and immune cells. They are activated by growth factors (e.g., EGF) as well as inflammatory (e.g., IFNα, IFNβ, IFNγ, IL-6, IL-23) and immunosuppressive (e.g., IL-10, IL-27, IL-35) cytokines. Once activated, JAKs phosphorylate specific members of the Signal Transducer and Activator of Transcription (STAT1-6) transcription factor family. Tyrosine phosphorylation allows the formation of STAT family homo- and hetero-dimers, which translocate to the nucleus where they induce the expression of hundreds of anti-viral, inflammatory, or immunosuppressive genes [154]. 

In solid cancers, STAT1 and STAT3 have pleiotropic roles during cancer development and in the establishment of immune responses. Type I (IFNα/β) and type II (IFNγ) interferons stimulate the formation of STAT1/STAT1 homodimers or STAT1/STAT2 heterodimers to induce inflammatory responses [155]. In contrast, the delayed formation of STAT1/STAT3 heterodimers downstream of IFN receptors negatively regulate STAT1-dependent inflammatory responses to induce immune suppression [156]. By the same token, multiple cytokines, including IL-6 and IL-10, induce the formation of STAT3/STAT3 homodimers, which increase the expression of genes that recruit and activate immunosuppressive macrophages (M2) and regulatory T cells [157]. Coupled with the observation that the STAT3 pathway also induces tumor growth, angiogenesis, and metastasis, several STAT3 inhibitors are in development to treat solid malignancies [158]. Indeed, we previously showed that an increased ratio of active STAT1/STAT3 in breast tumors is associated with improved immune surveillance, increased production of inflammatory cytokines, and enhanced sensitivity to immune checkpoint inhibitors in breast cancer [86].

The IFN/STAT1 pathway functions as a double-edged sword in cancer development [159]. Several studies have shown that it coordinates tumor-suppressive transcriptional responses, both in the tumor cells themselves and in cells within the tumor microenvironment through multiple mechanisms. IFN/STAT1 signaling potentiates cell cycle arrest and apoptosis in tumor cells, induces angiostatic responses, increases antigen processing and presentation by tumor cells, and primes the innate and adaptive immune cells to activate immune cell subsets (Th1/CTL; NK) that promote anti-tumor immunity. While STAT1 is required for immune surveillance, its chronic activation paradoxically potentiates and maintains tumor immune evasion by increasing the expression of immunosuppressive mediators (PD-L1, IDO1) [159]. Indeed, whereas STAT1 signaling in NK and T cells potentiates their effector functions [160,161], sustained STAT1 signaling in T cells protects them from NK-cell mediated cytotoxicity, preventing their elimination in inflamed tissues [162].

Several studies highlight this complex relationship between JAK/STAT1 signaling, tumor immunity, and sensitivity to immune checkpoint inhibitors in oncology. For example, whole exome and transcriptomic analysis of >1000 tumors treated with immune checkpoint inhibitors (across seven tumor types) showed that an elevated clonal tumor mutational burden, coupled with increased expression of CXCL9 (an IFN-inducible gene), is predictive of superior response [163]. In an independent study, macrophage-derived CXCL9 and CXCL10 were significantly elevated in response to immune checkpoint inhibition and were required to mount effective CTL-driven anti-tumor immune responses [164]. In high-grade serous ovarian carcinomas, elevated PD-L1 levels, indicative of increased STAT1 activation, is associated with elevated numbers of tumor-infiltrating lymphocytes and good outcome [165]. Finally, tumors with a stem-like phenotype possess decreased type I IFN/STAT1 signaling and are highly immunosuppressive, despite their high mutational burden [166]. Indeed, glioma stem cells evade immune surveillance by downregulating STAT1 expression at the epigenetic level [167]. 

In contrast, some studies point to IFN/STAT1 activation in the inferior response to kinase and/or immune checkpoint inhibition. For example, two neo-adjuvant clinical trials, including palbociclib (CDK4/6i) plus endocrine therapy, showed that increased IFN/STAT1 signaling in ER+ breast cancers were associated with elevated immune checkpoint levels, endocrine resistance, and poor outcome [168]. In pancreatic tumors, dinaciclib (a pan CDK inhibitor targeting CDK2/5/9) reversed immune suppression by inhibiting IFNγ-induced expression of immunosuppressive mediators, including IDO1 and PD-L1 [169]. Indeed, IFN/STAT1 activation in individual HER2+ breast cancers can exert opposing effects on their sensitivity to HER2 kinase inhibitors. Whereas Th1 cytokines, including IFNs, sensitized HER2+ tumors to lapatinib [170], trastuzumab increased PD-L1 expression in breast tumors, contributing to trastuzumab resistance [171]. 

Combined, these studies demonstrate a complex role for IFN/STAT1 signaling in tumor development and immune evasion, highlighting the need for further research to identify whether activation of this pathway will potentiate and/or suppress sensitivity to immune checkpoint inhibitors, alone or in combination with kinase inhibitors.

## 14. Kinase Inhibitors Target Immune Cells in the Tumor Microenvironment

Immune cells depend on multiple kinases to function. As such, kinase inhibitors used to treat cancer also directly regulate immune cell signaling and activity in the tumor microenvironment (reviewed in [172]), leading both to immune-related adverse events as well as improved sensitivity to immune-based therapies.

For example, skin inflammation is a prominent adverse effect of some EGFR inhibitors through off-target inhibition of Ste-10-like (STK10) serine/threonine kinase, leading to enhanced lymphocyte migration and secretion of IL-2 [173]. Moreover, immune checkpoint inhibitors are frequently associated with immune-related adverse events in cancer patients, and emerging evidence suggests that kinase inhibitors may be employed to modulate these toxicities. Indeed, the treatment of a melanoma patient experiencing anti-PD1 induced colitis with an mTOR inhibitor (sirolimus) could dampen systemic inflammatory responses and relieve this toxicity while sparing the anti-tumorigenic effects of PD-1 blockade [174].

Perturbation of signaling pathways in immune cells within the tumor microenvironment by kinase inhibitors also affects the efficacy of targeted and immune based-therapies, altering patient outcomes. Several RTKs, including Tyro3, Mer, and Axl (TAM-family), are expressed on APCs (macrophage and dendritic cells) and negatively regulate their activation and antigen presentation capabilities [175]. Inhibition of these TAM receptors on NK cells also leads to rejection of breast cancer metastasis in mouse models [176]. Given that intra-tumoral Tyro3, Mer, and Axl signaling contributes to tumor growth, immune evasion, drug resistance, proliferation, and metastasis [177,178,179], TAM RTKs represent an important drug candidate to simultaneously target both malignant cells and immune cells in order to enhance anti-tumor immunity. Similarly, other kinase inhibitors that induce tumor-intrinsic effects also target immune cells to simultaneously relieve immune suppression as part of their mechanism of action. While trametinib is used to treat many solid tumors, this MEK inhibitor also suppresses naïve CD8+ T cell priming and protects CD8+ T cells from chronic T cell receptor activation, leading to synergy with anti-PD-L1 inhibitors [115]. CDK4/6 inhibition also elicits increased antigen presentation in tumor cells and suppresses the proliferation of regulatory T cells to result in CTL-mediated anti-tumor immunity [60].

PI3K/AKT/mTOR signaling is critical in controlling immune cell function and has been linked to immunosuppressive immune cell function in the tumor milieu. PI3Kγ, which is expressed specifically in immune cells, is an important drug target for pan PI3K inhibitors in modulating anti-tumor immune responses. Indeed, PI3Kγ is required in Tregs and MDSCs to stimulate their infiltration into tumor tissue [180]. Moreover, PI3Kγ expression in macrophages negatively regulates the pro-inflammatory TLR4/NFkB signaling pathway while positively regulating IL-4 and C/EBPb signaling [181], polarizing macrophages toward immunosuppressive types. In line with this, PI3Kγ inhibitors can relieve macrophage-driven immunosuppression on T cells and synergize with PD-1 inhibitors to impede tumor growth [181]. Similarly, Mnk signaling in macrophages also increases their immunosuppressive properties [182]. On the other hand, dactolisib, a pan-specific PI3K inhibitor, reduced mRNA translation initiation in granulocytic MDSCs in a preclinical model of prostate cancer, inhibiting their immunosuppressive properties [183]. Indeed, dactolisib synergized with immune checkpoint blockade to induce durable tumoricidal responses in prostate cancer models [184]. Beyond this, mTOR signaling in multiple immune cell types contributes to immune suppression. For example, glioblastoma cells upregulate mTOR signaling in microglia, tissue-resident macrophages, which increases their immunosuppressive properties [185]. Furthermore, inhibition of mTOR signaling in T cells allows for their spontaneous activation into effector T cells, suggesting that this pathway is important for T cell tolerance. Indeed, mTOR function is required for the generation of regulatory T cells through metabolic reprogramming [186]. Similarly, mTOR activation increases fatty acid synthesis in dendritic cells, which indirectly reduces acetyl CoA pools, leading to reduced histone acetylation. This epigenetic reprogramming minimizes the ability of DCs to activate cytotoxic T lymphocytes [187].

While some studies indicate kinase inhibitors as promising agents to reverse immunosuppression in combination with cancer immunotherapy, others show that kinase inhibitors may dampen anti-tumor immune responses, potentially contributing to their lack of efficacy in cancer treatment [188]. For example, mTOR signaling plays a complex role in regulating NK cell function whereby mTORC1 signaling stimulates NK cell cytolytic function, whereas mTORC2 activity favors immunosuppressive NK cells [189]. Finally, mTORC2 deletion in macrophages stimulates a pro-inflammatory microenvironment that potentiates colitis-induced colon cancer [190]. These studies suggest that rapalogs (mTORC1-specific) and active site dual-specificity mTOR inhibitors (asTORi) may differentially impact the tumor immune microenvironment and, potentially, sensitivity to combination immunotherapies. Combined, these studies demonstrate that kinase inhibition impacts both tumor cells and immune cells in the tumor microenvironment to modulate treatment response and sensitivity to immunotherapy. Thus, it is critical to investigate the impact of individually targeted therapies on immune cells to maximize their ability to synergize with immunotherapies.

## 15. Conclusions

Collectively, a combination of kinase inhibitors and immunotherapy holds promise in cancer treatment. It remains to be seen whether kinase inhibitors combined with immunotherapy will be effective in different subtypes of cancers. Where there is a lack of response, its mechanisms should be investigated. The intrinsic kinase signaling pathways tumors employ to adapt and resist cancer immunotherapy need to be investigated. Where efficacy is seen, it must be established if a combination approach—over a single-agent approach—is preferred. In addition to duration, the order and timing of the combined approach, whether phased or sequential use of the drugs is as effective need to be tested [112]. Mechanistic studies that inform rational combination of kinase inhibitors and immunotherapeutic modalities for clinical trials are in need. Given all the evidence presented above, combinatorial use of kinase inhibitors and cancer immunotherapy may help to combat drug resistance and broaden responsiveness.

## Figures and Tables

**Figure 1 ijms-22-02608-f001:**
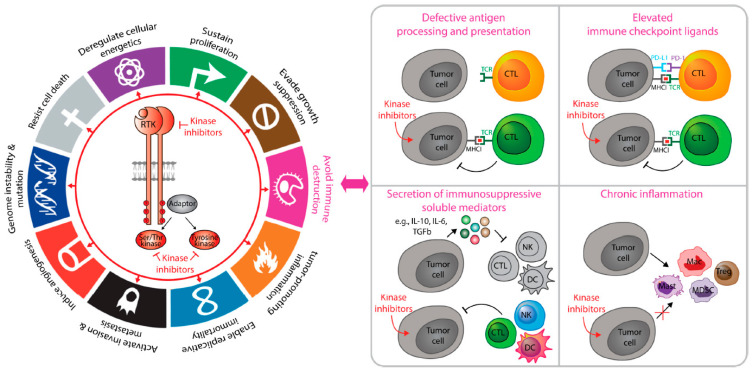
Pharmacological inhibition of kinase signaling relieves immunosuppression in solid malignancies. Deregulated activation of numerous kinases belonging to receptor tyrosine kinase, non-receptor tyrosine kinases, and serine/threonine kinase families induce gene expression changes that potentiate the growth and metastatic spread of solid tumors. In addition to influencing tumor cell-intrinsic processes that are essential for malignant progression, kinase signaling networks allow solid tumors to evade anti-tumor immune responses through multiple mechanisms. These include: decreasing antigen processing and presentation, increased secretion of immunosuppressive molecules, increased expression of immune checkpoint ligands, and stimulation of chronic inflammation. As such, pharmacological inhibitors of kinase signaling networks can relieve immune suppression and improve the sensitivity of solid malignancies to immune checkpoint inhibitors. Permission to use adapted figure elements originally published by Elsevier Press (Hallmarks of Cancer: The Next Generation) was obtained: License #5017191244575.

## Data Availability

No new data were created or analyzed in this study. Data sharing is not applicable to this article.

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
