# Peer review of "Clinical Potential of Kinase Inhibitors in Combination with Immune Checkpoint Inhibitors for the Treatment of Solid Tumors"

_ijms, 2021, doi:10.3390/ijms22052608_

Round 1
Reviewer 1 Report
Ahn and Ursini-Siegel have put forward an excellent review regarding the therapeutic synergy between immune checkpoint blockade and kinase inhibitors. The review provides an excellent summary of currently available therapeutics and ongoing combinatorial trials in the field.
A key strength of the paper is a well thought-out discussion of the potential immunosuppressive drawbacks of these kinase inhibition which are important considerations when applying these therapies. One addition consideration the authors might include is a discussion about the importance of IFNγ inducible chemokines in response to immune checkpoint blockade (PMID: 33508232, PMID: 31636098) and the very high likelihood that kinase inhibitors will block their production by immune and tumor cells (PMID: 26090665).
The authors may consider generating a summary diagram illustrating the immune consequences of combinatorial kinase inhibition and immune checkpoint blockade
Author Response
We thank the reviewer for his/her positive feedback regarding our review article. The reviewer makes an excellent point regarding including discussion on the importance of IFNg-inducible chemokines in inducing responsiveness to immune checkpoint inhibitors and the likelihood that kinase inhibitors will block their production. We have included this text and extended our discussion by incorporating a section address the importance of JAK/STAT signaling in controlling cancer immunosuppression, with a focus on STAT1 and STAT3. We discuss the pleiotropic roles of STAT1 signaling both in inducing anti-tumor immunity, in part but not limited to inducing said chemokine expression, and also in potentiating immune suppression by inducing expression of immunosuppressive ligands, such as IDO1 and PD-L1. The impact of kinase inhibitors on STAT1 activation, both in tumor cells and immune cells, and their consequence of perturbing STAT1 signaling on immune responses as well as sensitivity to immune checkpoint blockade is discussed. The new section including this text can be found on pages 11 and 12 of the revised manuscript and is entitled, “Kinase inhibitors impact JAK/STAT-mediated tumor immunity.”
The reviewer also requested a figure demonstrating the immune consequences of kinase inhibition of immune suppression. This is an excellent point and is now Figure 1 in the current manuscript.

Reviewer 2 Report
The review article on the Clinical potential of kinase inhibitors in combination with immune checkpoint inhibitors in tumor treatment is comprehensively described recent literature information, thoroughly covers various kinase inhibitors, immune checkpoint inhibitors on various cancer treatments, and FDA-approved kinase inhibitors that are currently in the treatment. This review will very beneficial, benefits upcoming researcher in the field. I have the following concerns in the review.
i) I would recommend authors to include an illustrative figure/rationale of a combination of kinase inhibtors and immune inhibitors in Breast cancer treatment/any solid tumor as an example in blocking tumor growth and enhance the immune cell response.
Author Response
We are grateful to this reviewer for his/her positive comments stating that the review “comprehensively described recent literature information, thoroughly covers various kinase inhibitors, immune checkpoint inhibitors on various cancer treatments, and FDA-approved kinase inhibitors that are currently in the treatment. This review will very beneficial, benefits upcoming researcher in the field.”
The reviewer requested that we include a figure demonstrating the immune consequences of kinase inhibition of immune suppression. This is an excellent point and is now Figure 1 in the current manuscript. We also note that in response to a request from the other reviewer, we have included a section on the importance of JAK/STAT signaling, focusing on the STAT1 and STAT3 pathways, in controlling immune suppression. We also describe recent literature discussing how kinase inhibitors impact IFN/STAT1 signaling to control immune surveillance and sensitivity to immune checkpoint inhibitors. This can be found on pages 11 and 12 of the revised manuscript.

Round 2
Reviewer 2 Report
The current version of the manuscript addresses my concerns raised in the previous version. I recommend the current version of the manuscript for publication.